RESEARCH ARTICLE *Mem Inst Oswaldo Cruz*, Rio de Janeiro, Vol. *121*: e250177, 2026   1|13

# Differential expression of miRNAs in Vero cells after Mayaro virus infection

**Juliana Santana de Curcio**[1/+]**, Lívia do Carmo Silva**[1]**, Evandro Novaes**[2]**,
Elisângela de Paula Silveira-Lacerda**[1/+]

[1]Universidade de Goiás, Departamento de Genética, Goiânia, GO, Brasil
[2]Universidade Federal de Lavras, Departamento de Biologia, Lavras, MG, Brasil

**BACKGROUND** Mayaro virus (MAYV) is an emerging arbovirus in the Americas that causes dengue-like illness with prolonged joint pain. Despite clinical relevance, its molecular interactions with host cells remain poorly understood.

**OBJECTIVES** Given the regulatory role of microRNAs (miRNAs) in viral infections, this study aimed to investigate miRNA expression profiles in MAYV-infected Vero cells.

**METHODS** Vero cells were infected with MAYV [multiplicity of infection (MOI) = 5] and analysed for viral replication, cell viability, and small RNA expression. Based on these parameters, the 24-h post-infection time point was selected for small RNA sequencing. Bioinformatic tools were used to identify differentially expressed miRNAs and predict their targets in *Homo sapiens* and the MAYV genome.

**FINDINGS** Among the 348 miRNAs identified, 46 were differentially expressed at 24 h (42 upregulated and four downregulated). Principal component analysis (PCA) indicated a clear separation between infected and control groups. *In silico* predictions of the targets of these miRNAs suggest potential associations with biological processes that may be relevant to virus-host interactions, such as immune response, programmed cell death pathways, viral replication, and persistence. Additionally, one miRNA detected in Vero cells was predicted to target a viral non-structural protein.

**MAIN CONCLUSIONS** Our findings indicate a potential dual role for host miRNAs during MAYV infection, involving both the modulation of host responses by the virus to enhance replication and a possible antiviral effect. While these interactions underscore the prospective relevance of miRNAs as biomarkers and therapeutic targets in arboviral infections, it is important to note that these conclusions are based solely on computational analyses. Therefore, they should be interpreted with caution until they are supported by further experimental validation.

Key words: diseases - Mayaro virus - host - small RNAs - cells

Arboviruses are a group of viral infections transmitted by arthropods, primarily mosquitoes. Among the most significant arboviruses are dengue (DENV), Zika (ZIKV), Chikungunya (CHIKV), and the Mayaro virus (MAYV).[1] MAYV has gained increasing attention in recent years due to the rising number of cases and its potential for outbreaks in endemic regions, particularly in the Americas.[2,3] This virus is classified as an alphavirus of the *Togaviridae* family and, although it primarily causes a febrile illness with symptoms similar to DENV or CHIKV such as fever, arthralgia, and skin rash, it has been associated with more severe and prolonged joint pain.[4,5,6,7]

Despite its growing clinical significance, the diagnosis of MAYV remains challenging, and currently, there are no specific antivirals or licensed vaccines available, making the infection an emerging public health concern. In this context, understanding the interactions between the virus and the host is essential to elucidate pathogenic mechanisms, identify immune evasion pathways, and explore potential therapeutic targets.[8,9] One promising field in this regard is the investigation of host factors, such as microRNAs (miRNAs), which play a crucial role in regulating immune responses and modulating viral replication. The study of these small non-coding RNAs can provide valuable insights into the molecular interactions between MAYV and host cells, aiding in the identification of diagnostic biomarkers and potential therapeutic strategies to control the infection and minimise its epidemiological impact.[10,11] Evidence from other arboviruses has demonstrated the crucial role of miRNAs in regulating viral infection and the host response. For example, a study analysing miRNA expression profiles in blood samples from patients with acute DENV infection identified 17 differentially expressed miRNAs that could be used to distinguish between mild cases and

Financial support: This work was funded by the Program of Academic Cooperation in National Defence (PROCAD)(Grant NR152019), CNPq, FAPEG.
+ Corresponding authors: julianadecurcio1@gmail.com | ⓘ https://orcid.org/0000-0003-2978-3689 / elacerda@ufg.br | ⓘ https://orcid.org/0000-0002-4143-9007

dengue haemorrhagic fever, highlighting their potential as diagnostic biomarkers. Additionally, the overexpression of miR-484 and miR-744 in Vero cells inhibited infection by all four DENV serotypes, suggesting a potential therapeutic target.[12,13] Similarly, ZIKV infection in mouse neurons reduced the expression of miR-155, miR-203, miR-29a, and miR-124-3p, miRNAs that regulate genes involved in antiviral immunity and neuroinflammation. Since these miRNAs are associated with brain lesions, their regulation could be explored for diagnostic purposes or neuroprotective therapies.[14] The differential regulation of 75 miRNAs, including those involved in JAK-STAT signalling and apoptosis key pathways for immune response has been identified in CHIKV infection.[15] These miRNAs could be explored as therapeutic targets to modulate the inflammatory response and reduce cellular damage.

Given the regulatory role of miRNAs in infections by DENV, ZIKV, and CHIKV, it is plausible that MAYV infection is also associated with a specific miRNA profile that can be further explored. Thus, the aim of this study is to investigate the differential expression of miRNAs in Vero cells following MAYV infection.

## MATERIALS AND METHODS

*Isolation of MAYV, cell culture* - The MAYV strain (BR/SJRP/LPV01/2015, GenBank accession number KT818520.1) was obtained from the National Reference Laboratory for Arboviruses in Campinas (Unicamp). The virus was propagated and used for infection experiments in Vero cells (ATCC CCL-81), a cell line known to be susceptible to arboviruses. Vero cells were cultured in Dulbecco's Modified Eagle Medium (DMEM) (Merck) supplemented with 10% foetal bovine serum (FBS) (ThermoFischer, scientific ), 1% penicillin (100 U/mL) and streptomycin (100 µg/mL) (Penicillin-Streptomycin Solution, Merck). Cells were maintained at 37ºC in a humidified 5% $CO_2$ incubator.

*Determination of infection time* - For infection, Vero cells were counted with trypan blue, and $1 \times 10^5$ cell/mL viable were seeded into a culture flask 25 cm² and infected with MAYV at a multiplicity of infection (MOI = 5, five viral particles per Vero cell). The viral concentration (number of viral copies) was determined based on the standard curve.[16] After 1 h of adsorption at 37ºC, the inoculum was removed, and the cells were washed with phosphate-buffered saline (PBS) (1X). Subsequently, the cells were maintained in DMEM containing 10% FBS, 1% penicillin (100 U/mL) and streptomycin (100 µg/mL), in 5% $CO_2$ at 37ºC. Viral infection was monitored by cytopathic effect (CPE) and confirmed by quantitative reverse transcription-polymerase chain reaction (qRT-PCR). The morphological characteristics of Vero cells were assessed at different time points (3, 4, 6, and 24 h) using brightfield microscopy (Leica DMIL LED) coupled with a 5 mp cmos usb digital camera, at 40 X magnification.

To evaluate viral replication, the supernatant was collected at each time point (3, 4, 6, and 24 h) and viral RNA was extracted using the MagMAX™ Viral/Pathogen Nucleic Acid Isolation Kit (Thermo Fisher Scientific), following the manufacturer's instructions. Briefly, samples were lysed, and nucleic acids were captured using magnetic beads, followed by multiple washing steps and elution in nuclease-free water.

The RNA was used in RT-qPCR, the reaction was performed on an AriaMX instrument (Agilent, Santa Clara, CA, USA). Oligonucleotide primers and probes targeting MAYV were obtained from IDT (Integrated DNA Technologies), as described by the Centres for Disease Control and Prevention (CDC). RT-qPCR was carried out using the GoTaq® Probe 1-Step RT-qPCR System (Promega, Madison, WI, USA) under the following conditions: 45ºC for 15 min for reverse transcription (1 cycle), followed by 95ºC for 2 min to inactivate the reverse transcriptase and activate GoTaq® DNA Polymerase (1 cycle). Then, 95ºC for 15 s, and 60ºC for 1 min for denaturation, annealing, and extension, for a total of 40 cycles.

The cell viability was assessed using the 3-(4,5-dimethylthiazol-2-yl)-2,5-diphenyltetrazolium bromide (MTT) (Invitrogen™) assay. The Vero cells were infected at a MOI = 5 (as described above) and incubated for 3, 4, 6, and 24 h in 96-well plates at a density of $1 \times 10^5$ cells per well. The experiment's negative control was performed with Vero cells only, without the addition of the MAYV. Subsequently 10 µL of MTT solution (5 mg/mL in PBS) was added per well and incubated for 3 h at 37ºC. The supernatant was then removed, and the formazan crystals were dissolved in 100 µL of DMSO (100% v/v) (Sigma). Absorbance was measured at 545 nm using a microplate spectrophotometer (Agilent Biotek Epoch). The experiment's cell viability was calculated as (absorbance of treatment/absorbance of negative control) × 100. In parallel, cell viability was also assessed using the trypan blue exclusion assay at different time points (3, 4, 6, and 24 h) post-infection. Cells were stained with trypan blue and counted using a Neubauer chamber to distinguish viable (unstained) from non-viable (blue-stained) cells. All experiments were performed in triplicate.

*RNA extraction and integrity analysis, next-generation sequencing (NGS)* - After determining the optimal experimental period (24 h), Vero cells were infected with a MAYV suspension at a MOI = 5 (as described above), using three independent biological replicates per group (n = 3). The control group consisted of non-infected Vero cells cultured under the same conditions. Cells were incubated at 37ºC and 5% $CO_2$ for 24 h. After the infection period, supernatant was removed, and the cells were washed twice with PBS 1×. Cells were detached using trypsin-EDTA (Sigma), centrifuged at 1000 × g for 5 min at 4ºC, and the cell pellet was used for RNA extraction. Total RNA was extracted using QIAzol Lysis Reagent (Qiagen), following the manufacturer's protocol. Briefly, cells were lysed in QIAzol, followed by phase separation with chloroform. RNA was precipitated with isopropanol, washed with 75% ethanol, and dissolved in nuclease-free water. To assess RNA quality and integrity, the following methods were used: 1% agarose gel electrophoresis, for visual inspection of RNA degradation; Nanodrop (Life Technologies), to evaluate purity (OD260/OD280 ratio); Qubit® 2.0 Fluorometer (Life Technologies), for precise RNA quantification ; 2100 BioAnalyzer system (Agilent), for RNA integrity determination (RIN score).

The extracted RNA was stored in the GenTegra™ RNA matrix (GenOne Biotechnologies) to ensure stability for transport and storage before sequencing. Small RNA libraries were prepared using the NEBNext® Multiplex Small RNA Library Prep Set for Illumina (Illumina Kit), according to the manufacturer's instructions. The protocol included: Ligation of adapters to the 3′ and 5′ ends of small RNAs; cDNA synthesis and PCR amplification; purification of small RNA fragments using polyacrylamide gel electrophoresis (PAGE). The libraries were sequenced by GenOne Soluções em Biotecnologia using the Illumina NovaSeq 6000 platform.

*Bioinformatics analyses for identification of miRNAs* - Initially, the quality of the raw sequencing data was assessed using the FastQC program. Low-quality reads and adapter sequences from library preparation and sequencing were then removed using Trimmomatic.[17] The trimmed sequences were then mapped to the *Chlorocebus sabaeus* genome (GEO Accession viewer) and further analysed using the miRDeep2 program.[18] The resulting mapping file was processed with the miRDeep2.pl script to identify pre-miRNA structures within the *C. sabaeus* genome. These FASTA sequences were then analysed using the RNAfold database (http://rna.tbi.univie.ac.at/cgi-bin/RNAWebSuite/RNAfold.cgi).[19]

*MiRNA differential expression analysis and prediction of target genes* - The expression levels of each miRNA in the samples were determined by counting the number of reads mapped to each miRNA using the quantifier.pl script from miRDeep2. This resulted in a count matrix, where each miRNA corresponded to a row and each library sample (infection and control, including replicates) represented a column. The count matrix was then subjected to statistical analysis of differential expression using a Negative Binomial Generalised Linear Model (GLM) via the R package DESeq2[20] as previously described Curcio et al.[21] Differentially expressed miRNAs were identified considering a 8' expression difference ($|\log2$ fold change$| > 3$), with significant adjusted p-value [false discovery rate (FDR) < 5%] and taking into account the statistical support from miRDeep score $\geq 4$ and significant p-randfold. The differentially expressed miRNAs in Vero cells post-infection were further analysed for their potential to induce gene silencing in the genomes of *Homo sapiens* and MAYV. To predict potential target genes, RNAhybrid v. 2[22] was used in conjunction with the 3' untranslated region (UTR) sequences of *H. sapiens* genes obtained using a custom Perl script from the Genome Reference Consortium Human Build 38 (GRCh38), downloaded from the NCBI (https://www.ncbi.nlm.nih.gov/datasets/genome/GCF_000001405.26/). Sequences from MAYV strains (NC_003417.1) available in the NCBI genome database (https://www.ncbi.nlm.nih.gov/genome/) was also retrieved for target prediction. Predicted miRNA targets in MAYV and *H. sapiens* were subsequently classified based on Gene Ontology (GO) annotations, considering categories related to biological processes, molecular functions, and cellular components. FastQ sequences were deposited on NCBI, under BioProject PRJNA1272649, upon publication. The metadata were made available to the reviewers through this provisional link (https://dataview.ncbi.nlm.nih.gov/object/PRJNA1272649?reviewer=28gcoul767dr84fi4m9qet4p8v).

## RESULTS

*Characterisation of MAYV infection in Vero cells: microscopy, PCR, and functional analyses* - We first assessed morphological changes at different time points post-infection. As shown in Fig. 1A, no evident CPE were observed in infected cells compared to the control group when using an MOI = 5. To quantify viral replication, real-time PCR was performed on culture supernatants, revealing a progressive increase in viral copies over 24 h (Fig. 1B). The viability of infected cells was monitored using trypan blue exclusion and MTT assays. Trypan blue staining indicated that cell viability remained around 90% in the early hours of infection, but declined to 82% after 24 h (Fig. 1C). Consistently, the MTT assay demonstrated a gradual decrease in metabolic activity, with viability measurements of 95% at 3 h, 87% at 4 h, and approximately 85% and 80% at 6 and 24 h post-infection, respectively (Fig. 1D). Based on these results, the 24-h post-infection time point was selected as the most appropriate for constructing small RNA libraries, as it ensures an optimal balance between viral replication and cell viability.

*Analysis of miRNA libraries and characterisation of their profiles* - The small RNA libraries from the control (n = 3) and infection (n = 3) groups yielded average sequence counts of 103.788.593 and 104.426.132, respectively, after read cleaning (data not shown). Following sequence mapping to the reference genome, miRNA precursors were identified using miRDeep2. A total of 348 miRNAs were identified with miRDeep2. Detailed results, available in Supplementary data (Table I), include miRNAs previously reported in other organisms, highlighting base complementarity within the seed region. The mature, complementary, and precursor sequences of the identified miRNAs are also provided.

After identifying miRNAs under both conditions, differential expression analyses were conducted. Principal component analysis (PCA) revealed distinct expression patterns between control and infected libraries, indicating that infection induces specific gene expression responses (Fig. 2A). Control and infection samples clustered separately, supporting the presence of infection-driven transcriptional changes. One replicate from the infection group exhibited a divergent distribution profile; however, this variability is consistent with previous reports of differential miRNA expression among biological replicates.[23]

A total of 348 miRNAs characterised across the libraries [Supplementary data (Table I)] were subjected to differential expression analysis using a ($\log2$ fold change of > 3). Among these, 46 miRNAs were differentially expressed between the two conditions [Supplementary data (Table II)]. Of these, 42 miRNAs were upregulated in response to MAYV infection in Vero cells, while 4 miRNAs were downregulated after 24 h of infection. The expression profiles of these 46 differentially expressed microRNAs are depicted in the heat map (Fig. 2B).

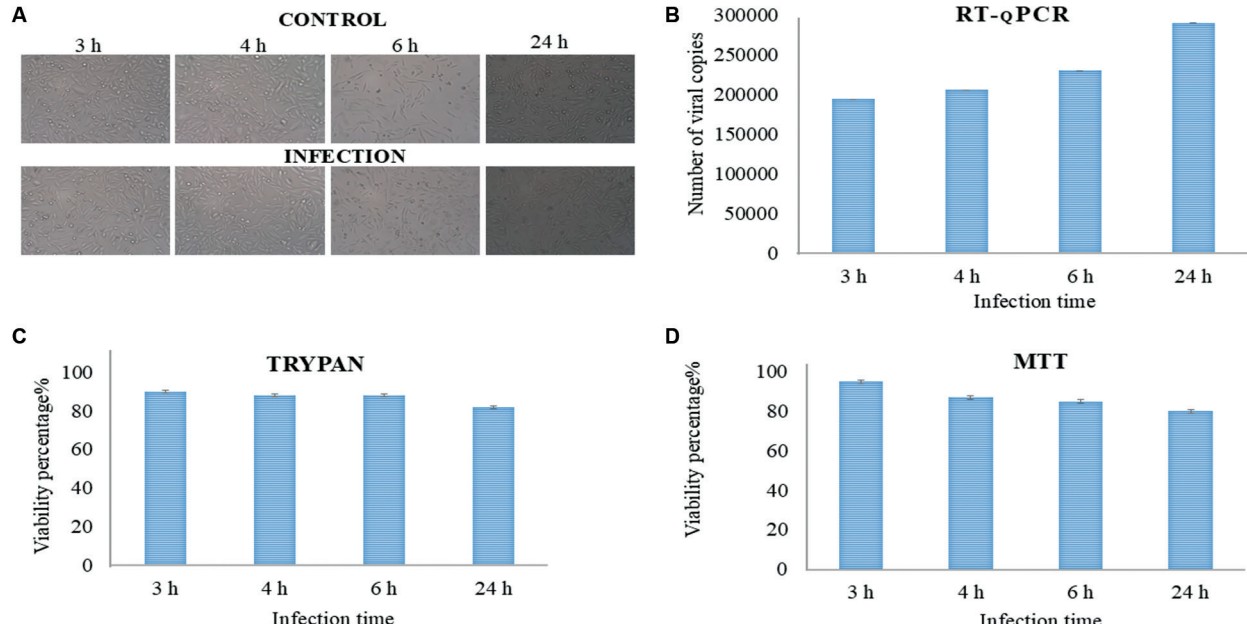

Fig. 1: analysis of Mayaro virus (MAYV) infection dynamics and cell viability in Vero cells. (A) Bright-field microscopy images of Vero cells infected with MAYV at a multiplicity of infection (MOI = 5) at different time points post-infection. (B) Quantification of viral replication in culture supernatants using reverse transcription quantitative polymerase chain reaction (RT-qPCR), with viral copy numbers determined based on a MAYV standard curve as described by Curcio et al.[21] (C) Cell viability assessment by trypan blue exclusion. (D) The 3-(4,5-dimethylthiazol-2-yl)-2,5-diphenyltetrazolium bromide (MTT) assay results showed the metabolic activity at different post-infection time points.

Differential expression analysis of host-derived miRNAs in Vero cells infected with MAYV revealed the regulation of several miRNAs associated with modulation of host biological processes, including immune response and viral persistence. Importantly, both the differentially expressed miRNAs and the affected biological pathways are of host origin, indicating that the virus may exploit the host's own regulatory networks to facilitate infection. The miRNA mml-let-7a-5p, encoded by the host genome, was found to be upregulated following infection, with Tollip (Toll-interacting protein) identified as a predicted target, suggesting suppression of Toll-like receptor signalling. Similarly, mml-miR-199a-1, another host miRNA, was induced in response to infection and exhibited multiple predicted targets related to host antiviral defence. Among these, APOBEC3H and SERINC4 both known host restriction factors that interfere with viral replication stood out. Additionally, the apoptotic regulators Bcl-2 and G0S2 were identified as potential targets, suggesting a role for mml-miR-199a-1 in modulating host cell death pathways to promote the survival of infected cells and support viral persistence.

The host miRNA hsa-miR-330-5p was also upregulated at 24 h post-infection, with ZBP1 (Z-DNA-binding protein 1) a cytosolic sensor of viral nucleic acids and a key activator of necroptosis predicted as a target, potentially impairing programmed cell death mechanisms. Lastly, hsa-miR-330-5p, another host-derived miRNA, showed increased expression after infection, with Macrophage Mannose Receptor 1 (MR), involved in the recognition of viral glycoproteins by host macrophages, predicted as a direct target. These results point to a potential mechanism by which MAYV could alter host

miRNA expression to evade immune responses and support viral replication and persistence a possibility that requires further experimental validation (Table I).

In relation to the host response, several host-derived miRNAs emerged as promising regulators. These miRNAs were expressed by host (Vero) cells and showed altered expression profiles following infection with MAYV. For example, differential expression analysis indicated that host miRNA, mmu-mir-615 was upregulated 24 h post-infection. Among its predicted targets, ZSWIM8, a substrate adaptor of the host CUL3 E3 ubiquitin ligase complex involved in the degradation of STAT2 and the regulation of type I interferon (IFN-I) responses stands out. Additionally, USP19, a host cytoplasmic deubiquitinase involved in protein quality control and immune modulation, was also predicted to be targeted by mmu-mir-615. In the context of MAYV infection, the host miRNA hsa-miR-330-5p was similarly upregulated 24 h after infection in Vero cells. This miRNA was predicted to target the host gene encoding the low-density lipoprotein receptor-related protein (LRP), which plays a role in viral entry via clathrin-mediated endocytosis. Its repression could impair viral attachment and internalisation, thereby influencing the infection (Table II). It is important to note that these proposed regulatory interactions and functional effects are hypotheses based solely on *in silico* predictions.

Furthermore, hsa-mir-31, a host-derived miRNA produced by Vero cells, was identified as the most strongly upregulated miRNA during MAYV infection, showing the highest level of induction 24 h post-infection (log2 fold change = 9.1) [Supplementary data (Table II)]. Notably, this miRNA exhibited base complementarity with

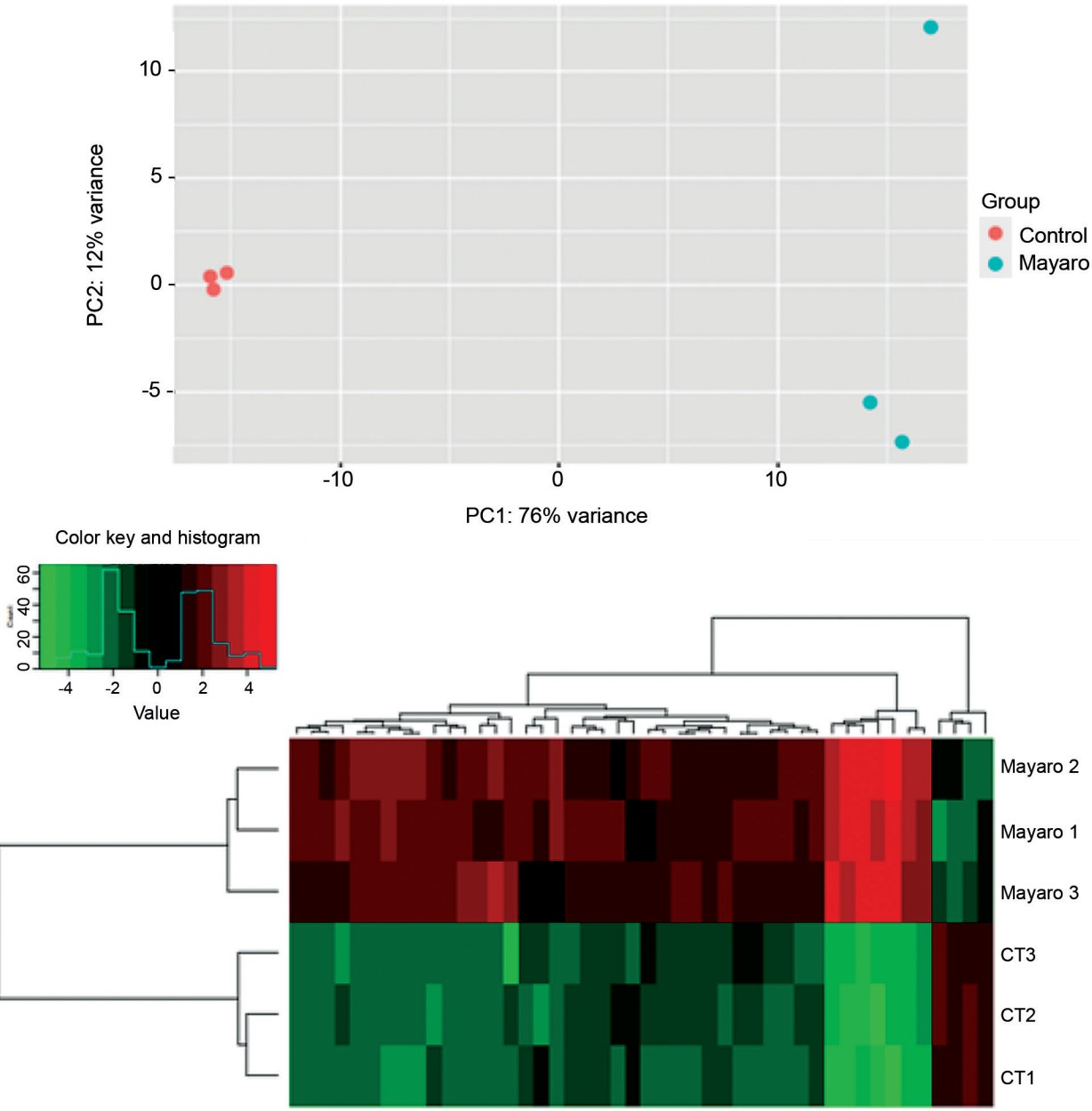

Fig. 2: differential expression pattern of miRNAs between the libraries.(A) Principal component analysis between libraries from control and infection conditions and (B) heat meap. CT: experimental control; Mayaro: Vero cells infected with the MAYV virus; 1, 2, 3: number of replicates.

a genomic region located in the viral 3′ UTR of the non-structural protein MAYVgp1 (ID: 935140), which is essential for viral replication (Fig. 3).

## DISCUSSION

The interaction between miRNAs and viruses occurs at multiple levels: while some host miRNAs exert antiviral effects by interfering with viral replication or the translation of viral proteins, others are hijacked by the virus to modulate signalling pathways that ultimately favour infection.[24] As highlighted by Bahojb Mahdavi et al.[25] this phenomenon has been observed in various clinically significant viral infections, including hepatitis B, hepatitis C, influenza, coronavirus, human immunodeficiency virus

(HIV), human papillomavirus, herpes simplex virus, Epstein-Barr virus, DENV, ZIKV, and Ebola viruses.

In this study the upregulation of mml-let-7a-5p was observed at 24 h post-infection suggests a potential involvement of this miRNA in modulating host immune signalling. Based *in silico* predictions, Tollip, a negative regulator of TLR and IL-1R pathways,[26] was identified as a putative target. Although Vero cells lack a functional type I interferon response, they retain components of TLR-mediated signalling.[27,28] Thus, this predicted interaction between mml-let-7a-5p and Tollip may hypothetically influence these pathways, potentially dampen inflammatory responses and create a more permissive environment for MAYV replication.

TABLE I

MiRNAs involved in modulating the response to viral infection persistence

| miRNAs | Selected target | Differential expression of miRNAs | Expected target expression |
|---|---|---|---|
| | Immune response | | |
| mml-let-7a-5p | Toll interacting protein (Tollip) | Upregulated | Downregulated |
| hsa-miR-330-5p | Macrophage mannose receptor 1(MCR1) | Upregulated | Downregulated |
| | Viral replication | | |
| mml-mir-199a-1 | Apolipoprotein B mRNA-editing enzyme, catalytic polypeptide-like 3 (APOBEC3H) | Upregulated | Downregulated |
| mml-mir-199a-1 | Serine incorporator 4 (SERINC4) | Upregulated | Downregulated |
| | Apoptotic process | | |
| mml-mir-199a-1 | G0/G1 switch 2 | Upregulated | Downregulated |
| mml-mir-199a-1 | Apoptosis regulator Bcl-2 | Upregulated | Downregulated |
| hsa-miR-330-5p | Z-DNA binding protein 1 (ZBP1 ) | Upregulated | Downregulated |

TABLE II

MiRNAs and their targets in the response the viral infection by host

| miRNAs | Selected target | Differential expression of miRNAs | Expected target expression |
|---|---|---|---|
| | Immune response | | |
| mmu-mir-615 | Zinc finger SWIM-type containing 8 (ZSWIM8) | Upregulated | Downregulated |
| | Viral replication | | |
| mmu-mir-615 | Ubiquitin specific peptidase 19 (USP19) | Upregulated | Downregulated |
| hsa-miR-330-5p | LDL receptor related protein 10 | Upregulated | Downregulated |
| hsa-mir-31 | Non-structural protein MAYVgp1 | Upregulated | Downregulated |

LDL: low-density lipoprotein; MAYV: Mayaro virus.

In this sense the overexpression of mml-miR-199a-1 following MAYV infection suggests a potential role of this miRNA in modulating host antiviral responses. *In silico* predictions identified APOBEC3H and SERINC4 proteins with documented antiviral activity against retroviruses such as HIV-1[29,30] as possible targets. The repression of these genes could hypothetically reduce intrinsic antiviral defences, favouring MAYV replication. Furthermore, the predicted modulation of Bcl-2 may delay apoptosis, allowing extended viral persistence. Interestingly, studies on DENV have shown that infection upregulates miR-15 and miR-16, enhancing apoptosis and viral dissemination through Bcl-2 modulation.[31] In contrast, the putative effects of mml-miR-199a-1 suggest that MAYV may adopt an opposite strategy, suppressing apoptosis to sustain replication and persistence within host cells.

Hsa-miR-330-5p is predicted, through *silico* analysis, to regulate two main targets during MAYV infection: the ZBP1 protein and the mannose receptor 1 (MR). ZBP1 activates necroptosis as an antiviral defence mechanism, but its silencing by miRNA-287 may suppress this pro-cess, favouring viral replication. This mechanism resembles immune evasion strategies observed in viruses of the *Herpesviridae* family, such as HSV-1 and MCMV, which produce RHIM domain-containing proteins capable of inhibiting the ZBP1-RIPK3 interaction.[32,33] Meanwhile, MR facilitates viral entry into macrophages, and its modulation by miRNA-287 may affect both infection and the host immune response, as described in dengue virus infections, where MR is induced by IL-4 and IL-13 and increases cellular susceptibility.[34] Thus, hsa-miR-330-5p may act as a post-transcriptional regulator, contributing to viral adaptation to the cellular environment and the host immune response.

Since Vero cells lack the ability to produce type IFN-I due to deletions in the IFN gene cluster, the regulatory effect of mmu-mir-615 on ZSWIM8 observed in this study likely occurs independently of endogenous IFN-I signalling. Our *in silico* prediction indicates that mmu-mir-615 targets ZSWIM8, a substrate adaptor of the CUL3 E3 ubiquitin ligase complex. In normal IFN-competent cells, repression of ZSWIM8 could

stabilise STAT2 by preventing its ubiquitin-mediated degradation, sustaining IFN-I signalling and enhancing antiviral responses. However, in Vero cells, this mechanism might instead reflect a residual or alternative regulatory pathway affecting STAT2 stability, rather than a direct modulation of IFN signalling. This contrasts with viral immune evasion strategies such as those described for ZIKV, whose NS5 protein recruits the ZSWIM8-CUL3 complex to degrade STAT2 and suppress the host antiviral response.[35] Therefore, the upregulation of mmu-mir-615 observed here may suggest a host-derived attempt to preserve components of antiviral defence even in the absence of functional IFN production.

**Target:** *MAYVgp1_3UTR::NC_003417.1:7389–7590(+)*
length: 201
**MiRNA:** *hsa-mir-31*
length: 23

mfe: −33.2 kcal/mol
p–value: 1.000000e+00

*Position: 159*

```
target 5' U      G          CG      C 3'
          GCAGC    GUUAGCA  CUUGCC
          UGUCG    CGGUCGU  GAACGG
miRNA  3'      AUA        A        A 5'
```

Fig. 3: base complementarity between the 3' untranslated region of Mayaro virus (MAYV) and differentially regulated miRNA. The RNAhybrid program was used to determine the base complementarity between the potential miRNA targets.

We observed the modulation of mmu-mir-615 expression in Vero cells infected with MAYV, suggesting a potential role for this miRNA in regulating host factors that affect viral replication. Among its predicted targets, USP19 a cytoplasmic deubiquitinase (DUB), involved in protein quality control, stress responses, and immune regulation emerges as a candidate with possible antiviral relevance. By analogy to other DUBs such as USP46 and USP12, which modulate gene expression during Epstein-Barr virus (EBV)-mediated transformation.[36] USP19 may similarly support MAYV replication by stabilising host or viral proteins required for the viral life cycle. Thus, mmu-mir-615 mediated repression of USP19 could modify the cellular environment, reducing permissiveness to viral propagation, either by promoting degradation of pro-viral factors or by enhancing stress-related antiviral responses. These findings suggest a potential host-derived mechanism that limits MAYV infection and may represent a promising target for future antiviral strategies.

The upregulation of hsa-miR-330-5p at 24 h post-infection in Vero cells suggests a potential regulatory role during MAYV infection. Our prediction indicates that miRNA-287 targets the LRP, which has been implicated as an entry receptor for several viruses, including Crimean-Congo haemorrhagic fever virus (CCHFV).[37] Although Vero cells lack a fully competent interferon response, the downregulation of LRP by miRNA-287 may represent a host attempt to restrict viral attachment and internalisation, thereby limiting MAYV propagation. This speculative mechanism raises the possibility that hsa-miR-330-5p contributes to an antiviral response by modulating receptor availability at the cell surface.

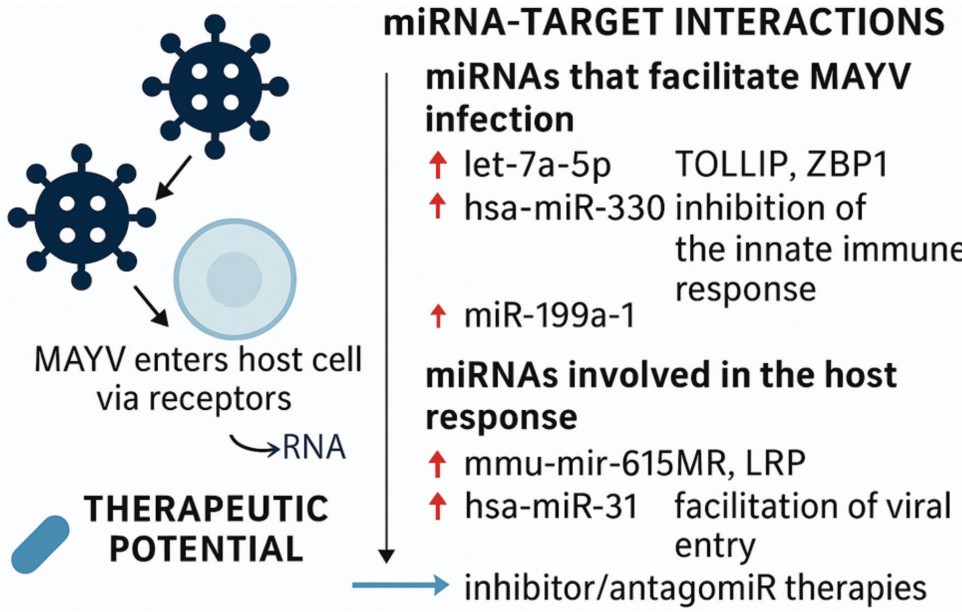

Fig. 4: schematic representation of *in silico* predictions of the role of host microRNAs in modulating cellular responses during Mayaro virus (MAYV) infection. Predicted interactions suggest that miRNAs such as let-7a-5p and miR-199a-1 may suppress key antiviral and immune-related proteins, potentially facilitating viral replication and immune evasion. Additionally, hsa-miR-31 is predicted to directly bind to the 3' untranslated region of MAYV ribonucleic acid (RNA), possibly repressing the translation of non-structural polyprotein and impacting viral replication. These predicted interactions highlight potential targets for therapeutic intervention against emerging arboviruses.

Understanding viral entry mechanisms underscores the need for precise and timely diagnostic tools to detect arboviral infections. Current diagnosis combines serological, molecular, and virological approaches to ensure accuracy and speed. Serological assays such as enzyme-linked immunosorbent assay (ELISA) are useful for retrospective and epidemiological analyses,[38] while molecular methods like RT-PCR enable sensitive detection of viral RNA during the acute phase.[39] Although traditional viral isolation remains valuable for surveillance and characterisation,[40] emerging technologies such as biosensors and NGS offer promising advances for rapid detection and genomic monitoring.[41]

Recent advances highlight miRNAs as key regulators of arbovirus replication and pathogenicity. In our study, hsa-mir-31 was upregulated in MAYV-infected Vero cells and showed complementarity to a region encoding non-structural polyproteins essential for viral replication, suggesting a potential post-transcriptional control mechanism. This interaction could influence replication efficiency and virion formation, reinforcing the idea that host miRNAs may serve as antiviral modulators or therapeutic targets. Although our findings are based on Vero cells, further studies are needed to confirm whether MAYV modulates hsa-mir-31 expression to favour or repress the infection. Similar host-virus interactions have been described for other arboviruses, where host miRNAs such as miR-142-3p,[42] let-7c[43] and others modulate viral RNA stability and replication.[44] The convergence of these findings highlights the importance of miRNAs both as promising tools for developing therapeutic strategies aimed at restricting viral replication and as potential diagnostic biomarkers capable of indicating the presence and progression of viral infection. Unlike conventional methods, miRNA-based approaches may allow early and specific detection of infection, as well as provide a route for interventions that directly modulate the viral life cycle and host immune response. Thus, incorporating miRNA analysis into the diagnostic and therapeutic arsenal could significantly contribute to the control of arboviral diseases, including those caused by MAYV (Fig. 4).

## ACKNOWLEDGEMENTS

To National Reference Laboratory for Arboviruses in Campinas (Unicamp).

## AUTHORS' CONTRIBUTION

CJS - conception and design, analysis and interpretation of the data, investigation, methodology, writing - original draft, writing - review & editing; CLS - methodology, writing - original draft, writing - review & editing; NE - data curation, formal analysis, writing - review & editing; SLEP - funding acquisition, project administration, resources, investigation, writing - review & editing. All authors agree to be accountable for all aspects of the work. The author reports no financial interests or other conflicts of interest.

## DATA AVAILABILITY

The datasets generated during the current study are available in the: https://dataview.ncbi.nlm.nih.gov/object/PRJNA1272649?reviewer=28gcoul767dr84fi4m9qet4p8v.

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

# OPEN PEER REVIEW

Memórias do IOC thanks the anonymous reviewers for their contribution to the peer review of this work.

**FIRST REVIEW ROUND**

REVIEWERS' COMMENTS

**REVIEWER #1**

In this manuscript, Curcio et al. address an important gap in our understanding of host–virus interactions for Mayaro virus (MAYV). The use of small RNA sequencing to profile host miRNA responses is timely, and the findings may help set the stage for further mechanistic work. Overall, the study is relevant and potentially impactful, but several issues limit its current strength.

Major Comments

The choice of Vero cells as the host model is a significant limitation. These cells lack a functional interferon response, which is central to host–virus interactions and miRNA regulation. While I understand their convenience, the absence of this pathway must be acknowledged more explicitly in the manuscript, and the interpretation of results should be carefully framed.

The multiplicity of infection is described as "MOI 1:5." This is confusing, as MOI is conventionally expressed as a single number (e.g., MOI = 5). It needs to be clarified and consistent throughout the manuscript.

Only three replicates were used, and one infection replicate showed divergent clustering in PCA. While this is not unusual in sequencing studies, the variability deserves more careful discussion, and ideally, additional replicates should be included to strengthen confidence in the findings.

The use of a log2 fold-change > 3 as the threshold for calling differential expression is very stringent. This could exclude biologically relevant changes. Please provide justification for this cutoff and clarify whether multiple testing correction (e.g., FDR) was applied.

The conclusions about specific miRNAs (e.g., miR-35 targeting the viral 3′UTR, miR-199a-1 regulating APOBEC3H) are based solely on in silico predictions. Without experimental validation, these results should be framed as hypotheses rather than definitive findings. At a minimum, the discussion should be reworded to avoid overinterpretation.

There is some inconsistency in the miRNA naming (e.g., mml-, cae-, NC_023643.1_14184). It is not clear whether these are species-specific annotations or database artifacts. Please clarify and use consistent nomenclature throughout.

Minor Comments

The abstract could be tightened to avoid overstatements. For instance, "miR-35 showed the highest upregulation and complementarity to the 3′UTR of MAYVgp1" should be presented more cautiously, since no direct validation was performed.

Some sections are wordy or grammatically awkward. A careful round of editing would improve readability.

Overall Recommendation

I recommend a major revision. The dataset is potentially valuable, but the authors need to clarify methodological details, temper their conclusions, and better acknowledge the limitations of the model system. If they can address these concerns, the work will be a useful contribution to the field of arbovirology and host–pathogen interactions.

AUTHORS' RESPONSE TO THE REVIEWERS

To the Responsible Editor
Memórias do Instituto Oswaldo Cruz
Subject: Response to Manuscript ID MIOC-2025-0177, entitled "Differential expression of miRNAs in Vero cells after Mayaro virus infection"
From: Dr. Ana Carolina Vicente

Reviewer comments:
• Reviewer Comments
1-In this manuscript, Curcio et al. address an important gap in our understanding of host–virus interactions for Mayaro virus (MAYV). The use of small RNA sequencing to profile host miRNA responses is timely, and the findings may help set the stage for further mechanistic work. Overall, the study is relevant and potentially impactful, but several issues limit its current strength.

Major Comments

The choice of Vero cells as the host model is a significant limitation. These cells lack a functional interferon response, which is central to host–virus interactions and miRNA regulation. While I understand their convenience, the absence of this pathway must be acknowledged more explicitly in the manuscript, and the interpretation of results should be carefully framed.

Response : We thank the reviewer for this insightful comment. We agree that the lack of a functional interferon response in Vero cells is a major limitation, as the interferon pathway is central to host–virus interactions and miRNA regulation. In the revised manuscript (Discussion, lines 347-349), we explicitly acknowledge this limitation and have carefully reframed our interpretation of the results.

At the same time, we emphasize that Vero cells are a widely used experimental model for arbovirus infection studies, including Mayaro virus (MAYV), due to their high permissiveness to viral replication (Esposito et al., 2019 ; Mendonça et al., 2023). Therefore, while our results must be interpreted with caution, they provide an initial exploratory overview of miRNA responses in a system that is susceptible to infection and may represent potential mechanisms occurring in the host. We also highlight that future studies using interferon-competent primary or immortalized human cells will be essential to validate and extend these observations.

2- The multiplicity of infection is described as "MOI 1:5." This is confusing, as MOI is conventionally expressed as a single number (e.g., MOI = 5). It needs to be clarified and consistent throughout the manuscript.

The correction has been made

3- 1-Only three replicates were used, and one infection replicate showed divergent clustering in PCA. While this is not unusual in sequencing studies, the variability deserves more careful discussion, and ideally, additional replicates should be included to strengthen confidence in the findings?

Response: Three biological replicates are commonly used and considered adequate in RNA-Seq experiments. It is important to note that, contrary to the reviewer's impression, the three biological replicates do not show divergent clustering in the PCA. The apparent separation of one replicate occurs only along the second principal component (PC2), which explains 12% of the total variance, while all replicates cluster closely along PC1, which accounts for 76% of the variance. Therefore, when the PCA is interpreted in light of the variance explained by each component, the replicates are, in fact, tightly grouped and show a consistent transcriptional profile

4- The use of a log2 fold-change > 3 as the threshold for calling differential expression is very stringent. This could exclude biologically relevant changes. Please provide justification for this cutoff and clarify whether multiple testing correction (e.g., FDR) was applied?

Response :The threshold of |log2 fold-change| > 3 is indeed stringent. Because our analysis initially identified a large number of significantly differentially expressed miRNAs (209 at FDR < 5%), we chose to focus on those showing the most pronounced expression differences. In this exploratory study on miRNA responses to Mayaro virus infection, we prioritized minimizing false positives over capturing every possible change, aiming to identify the most robust expression signals for downstream interpretation.

Yes, multiple testing correction was applied using the False Discovery Rate (FDR) method as implemented in DESeq2. We thank the reviewer for noting this omission. The description in the Materials and Methods section has now been corrected.

Lines 202-203 now read:

Differentially expressed miRNAs were identified considering a 8× expression difference (|log2 fold change| > 3), with significant adjusted p-value (FDR < 5%) and taking into account the statistical support from miRDeep score ≥ 4 and significant p-randfold."

5- The conclusions about specific miRNAs (e.g., miR-35 targeting the viral 3′UTR, miR-199a-1 regulating APOBEC3H) are based solely on in silico predictions. Without experimental validation, these results should be framed as hypotheses rather than definitive findings. At a minimum, the discussion should be reworded to avoid overinterpretation.

Response : The discussion has been revised taking into account the requested corrections

6- There is some inconsistency in the miRNA naming (e.g., mml-, cae-, NC_023643.1_14184). It is not clear whether these are species-specific annotations or database artifacts. Please clarify and use consistent nomenclature throughout.

Response: The names of the miRNAs have been corrected.

Minor Comments

7- The abstract could be tightened to avoid overstatements. For instance, "miR-35 showed the highest upregulation and complementarity to the 3′UTR of MAYVgp1" should be presented more cautiously, since no direct validation was performed.

Some sections are wordy or grammatically awkward. A careful round of editing would improve readability.

Response: The abstract has been revised to avoid overstatements. Specifically, the sentence regarding miR-35 now reads more cautiously, reflecting that no direct experimental validation was performed.

The manuscript has undergone careful editing to improve readability, reduce wordiness, and correct grammatical issues throughout.

8- Overall Recommendation

I recommend a major revision. The dataset is potentially valuable, but the authors need to clarify methodological details, temper their conclusions, and better acknowledge the limitations of the model system. If they can address these concerns, the work will be a useful contribution to the field of arbovirology and host–pathogen interactions.

Response : We thank the reviewer for the constructive feedback. We have addressed the concerns by clarifying methodological details, tempering the conclusions, and more clearly acknowledging the limitations of the Vero cell model. We believe these revisions strengthen the manuscript and improve its contribution to the field of arbovirology and host–pathogen interactions.

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

## SECOND REVIEW ROUND

### REVIEWERS' COMMENTS

### REVIEWER #1

The authors have satisfactorily addressed the reviewers' prior comments. The manuscript is scientifically sound and may be accepted for publication following the incorporation of the minor revisions listed below.

Lines 112–113:

Please revise the sentence to read: "The viral concentration (number of viral copies) was determined based on the standard curve as described by Curcio et al. or as described previously (reference)." Ensure that the reference is cited in accordance with the journal's citation format.

Page 33, Figure 1 legend:

The phrase "MAYV standard curve as described by (21)" should be replaced with "MAYV standard curve as described previously [reference according to journal format]."

Verb tense consistency:

Verify that all procedural descriptions are written in the past tense (e.g., "was determined" rather than "is determined") to maintain consistency throughout the Methods section.

### AUTHORS' RESPONSE TO THE REVIEWERS

Response to the Editor and Reviewer
Manuscript ID: MIOC-2025-0177.R1
Title: Differential expression of miRNAs in Vero cells after Mayaro virus infection
Journal: Memórias do Instituto Oswaldo Cruz

Dear Dr. Ana Carolina Vicente,

We sincerely appreciate the reviewer's positive evaluation and their thoughtful comments, which have helped us further improve the quality and clarity of our manuscript. Below, we provide a detailed response to each point raised and indicate where the revisions were made in the manuscript.

Reviewer #1 Comments and Author Responses:

Comment: Please revise the sentence on Lines 112–113 to ensure citation format compliance.

Response: The sentence has been corrected as requested to: "The viral concentration (number of viral copies) was determined based on the standard curve as described by Curcio et al. 2022. The citation formatting was updated according to journal guidelines.

Location: Methods Section — Lines 112–113

Comment: Replace "MAYV standard curve as described by (21)" in the Figure 1 legend.

Response: The phrase has been replaced with: "MAYV standard curve as described previously by Curcio et al.  2022.

Location: Page 33 — Figure 1 Legend

Comment: Ensure consistency of past tense in procedural descriptions.

Response: All procedural sentences in the Methods section were reviewed and revised to maintain past tense consistently.

We hope that the revisions meet the expectations of the Editor and Reviewer. We remain at your disposal for any further modifications that may be needed.

Sincerely,
Juliana Santana de Curcio

## THIRD REVIEW ROUND

REVIEWERS' COMMENTS

**REVIEWER #1**

No comments.

