## [Reviewer Report · FIRST REVIEW ROUND - REVIEWERS COMMENTS]

## REVIEWER #1

In this manuscript, Curcio et al. address an important gap in our understanding of host–virus interactions for Mayaro virus (MAYV). The use of small RNA sequencing to profile host miRNA responses is timely, and the findings may help set the stage for further mechanistic work. Overall, the study is relevant and potentially impactful, but several issues limit its current strength.

**Major Comments**

The choice of Vero cells as the host model is a significant limitation. These cells lack a functional interferon response, which is central to host–virus interactions and miRNA regulation. While I understand their convenience, the absence of this pathway must be acknowledged more explicitly in the manuscript, and the interpretation of results should be carefully framed.

The multiplicity of infection is described as "MOI 1:5." This is confusing, as MOI is conventionally expressed as a single number (e.g., MOI = 5). It needs to be clarified and consistent throughout the manuscript.

Only three replicates were used, and one infection replicate showed divergent clustering in PCA. While this is not unusual in sequencing studies, the variability deserves more careful discussion, and ideally, additional replicates should be included to strengthen confidence in the findings.

The use of a log2 fold-change > 3 as the threshold for calling differential expression is very stringent. This could exclude biologically relevant changes. Please provide justification for this cutoff and clarify whether multiple testing correction (e.g., FDR) was applied.

The conclusions about specific miRNAs (e.g., miR-35 targeting the viral 3′UTR, miR-199a-1 regulating APOBEC3H) are based solely on in silico predictions. Without experimental validation, these results should be framed as hypotheses rather than definitive findings. At a minimum, the discussion should be reworded to avoid overinterpretation.

There is some inconsistency in the miRNA naming (e.g., mml-, cae-, NC_023643.1_14184). It is not clear whether these are species-specific annotations or database artifacts. Please clarify and use consistent nomenclature throughout.

**Minor Comments**

The abstract could be tightened to avoid overstatements. For instance, "miR-35 showed the highest upregulation and complementarity to the 3′UTR of MAYVgp1" should be presented more cautiously, since no direct validation was performed.

Some sections are wordy or grammatically awkward. A careful round of editing would improve readability.

**Overall Recommendation**

I recommend a major revision. The dataset is potentially valuable, but the authors need to clarify methodological details, temper their conclusions, and better acknowledge the limitations of the model system. If they can address these concerns, the work will be a useful contribution to the field of arbovirology and host–pathogen interactions.

## AUTHORS' RESPONSE TO THE REVIEWERS

**To the Responsible Editor**

Memórias do Instituto Oswaldo Cruz

Subject: Response to Manuscript ID MIOC-2025-0177, entitled "Differential expression of miRNAs in Vero cells after Mayaro virus infection"

From: Dr. Ana Carolina Vicente

**Reviewer Comments**

**1- In this manuscript, Curcio et al. address an important gap in our understanding of host–virus interactions for Mayaro virus (MAYV). The use of small RNA sequencing to profile host miRNA responses is timely, and the findings may help set the stage for further mechanistic work. Overall, the study is relevant and potentially impactful, but several issues limit its current strength.**

**Major Comments**

**The choice of Vero cells as the host model is a significant limitation. These cells lack a functional interferon response, which is central to host–virus interactions and miRNA regulation. While I understand their convenience, the absence of this pathway must be acknowledged more explicitly in the manuscript, and the interpretation of results should be carefully framed.**

*Response:* We thank the reviewer for this insightful comment. We agree that the lack of a functional interferon response in Vero cells is a major limitation, as the interferon pathway is central to host–virus interactions and miRNA regulation. In the revised manuscript (Discussion, lines 347-349), we explicitly acknowledge this limitation and have carefully reframed our interpretation of the results.

At the same time, we emphasize that Vero cells are a widely used experimental model for arbovirus infection studies, including Mayaro virus (MAYV), due to their high permissiveness to viral replication (Esposito et al., 2019; Mendonça et al., 2023). Therefore, while our results must be interpreted with caution, they provide an initial exploratory overview of miRNA responses in a system that is susceptible to infection and may represent potential mechanisms occurring in the host. We also highlight that future studies using interferon-competent primary or immortalized human cells will be essential to validate and extend these observations.

**2- The multiplicity of infection is described as "MOI 1:5." This is confusing, as MOI is conventionally expressed as a single number (e.g., MOI = 5). It needs to be clarified and consistent throughout the manuscript.**

*Response:* The correction has been made.

**3- Only three replicates were used, and one infection replicate showed divergent clustering in PCA. While this is not unusual in sequencing studies, the variability deserves more careful discussion, and ideally, additional replicates should be included to strengthen confidence in the findings.**

*Response:* Three biological replicates are commonly used and considered adequate in RNA-Seq experiments. It is important to note that, contrary to the reviewer's impression, the three biological replicates do not show divergent clustering in the PCA. The apparent separation of one replicate occurs only along the second principal component (PC2), which explains 12% of the total variance, while all replicates cluster closely along PC1, which accounts for 76% of the variance. Therefore, when the PCA is interpreted in light of the variance explained by each component, the replicates are, in fact, tightly grouped and show a consistent transcriptional profile.

**4- The use of a log2 fold-change > 3 as the threshold for calling differential expression is very stringent. This could exclude biologically relevant changes. Please provide justification for this cutoff and clarify whether multiple testing correction (e.g., FDR) was applied.**

*Response:* The threshold of |log2 fold-change| > 3 is indeed stringent. Because our analysis initially identified a large number of significantly differentially expressed miRNAs (209 at FDR < 5%), we chose to focus on those showing the most pronounced expression differences. In this exploratory study on miRNA responses to Mayaro virus infection, we prioritized minimizing false positives over capturing every possible change, aiming to identify the most robust expression signals for downstream interpretation.

Yes, multiple testing correction was applied using the False Discovery Rate (FDR) method as implemented in DESeq2. We thank the reviewer for noting this omission. The description in the Materials and Methods section has now been corrected.

Lines 202-203 now read: "Differentially expressed miRNAs were identified considering a 8× expression difference (|log2 fold change| > 3), with significant adjusted p-value (FDR < 5%) and taking into account the statistical support from miRDeep score ≥ 4 and significant p-randfold."

**5- The conclusions about specific miRNAs (e.g., miR-35 targeting the viral 3′UTR, miR-199a-1 regulating APOBEC3H) are based solely on in silico predictions. Without experimental validation, these results should be framed as hypotheses rather than definitive findings. At a minimum, the discussion should be reworded to avoid overinterpretation.**

*Response:* The discussion has been revised taking into account the requested corrections.

**6- There is some inconsistency in the miRNA naming (e.g., mml-, cae-, NC_023643.1_14184). It is not clear whether these are species-specific annotations or database artifacts. Please clarify and use consistent nomenclature throughout.**

*Response:* The names of the miRNAs have been corrected.

**Minor Comments**

**7- The abstract could be tightened to avoid overstatements. For instance, "miR-35 showed the highest upregulation and complementarity to the 3′UTR of MAYVgp1" should be presented more cautiously, since no direct validation was performed.**

**Some sections are wordy or grammatically awkward. A careful round of editing would improve readability.**

*Response:* The abstract has been revised to avoid overstatements. Specifically, the sentence regarding miR-35 now reads more cautiously, reflecting that no direct experimental validation was performed. The manuscript has undergone careful editing to improve readability, reduce wordiness, and correct grammatical issues throughout.

**8- Overall Recommendation**

**I recommend a major revision. The dataset is potentially valuable, but the authors need to clarify methodological details, temper their conclusions, and better acknowledge the limitations of the model system. If they can address these concerns, the work will be a useful contribution to the field of arbovirology and host–pathogen interactions.**

*Response:* We thank the reviewer for the constructive feedback. We have addressed the concerns by clarifying methodological details, tempering the conclusions, and more clearly acknowledging the limitations of the Vero cell model. We believe these revisions strengthen the manuscript and improve its contribution to the field of arbovirology and host–pathogen interactions.

**References**

Esposito DL, Di Caro A, Capobianchi MR, et al. Infection and pathogenesis of Mayaro virus in vertebrate and invertebrate hosts: a review. Viruses. 2019;11(5):447. doi:10.3390/v11050447.

Mendonça DC, Reis EVS, Arias NEC, et al. A study of the MAYV replication cycle: Correlation between the kinetics of viral multiplication and viral morphogenesis. Virus Res. 2023;323:199002. doi:10.1016/j.virusres.2022.199002.

---

## [Reviewer Report · REVIEWERS COMMENTS]

## REVIEWER #1

The authors have satisfactorily addressed the reviewers' prior comments. The manuscript is scientifically sound and may be accepted for publication following the incorporation of the minor revisions listed below.

Lines 112–113: Please revise the sentence to read: "The viral concentration (number of viral copies) was determined based on the standard curve as described by Curcio et al. or as described previously (reference)." Ensure that the reference is cited in accordance with the journal's citation format.

Page 33, Figure 1 legend: The phrase "MAYV standard curve as described by (21)" should be replaced with "MAYV standard curve as described previously [reference according to journal format]."

Verb tense consistency: Verify that all procedural descriptions are written in the past tense (e.g., "was determined" rather than "is determined") to maintain consistency throughout the Methods section.

## AUTHORS' RESPONSE TO THE REVIEWERS

**Response to the Editor and Reviewer**

Manuscript ID: MIOC-2025-0177.R1

Title: Differential expression of miRNAs in Vero cells after Mayaro virus infection

Handling Editor: Dr. Ana Carolina Vicente

Journal: Memórias do Instituto Oswaldo Cruz

Dear Dr. Ana Carolina Vicente,

We sincerely appreciate the reviewer's positive evaluation and their thoughtful comments, which have helped us further improve the quality and clarity of our manuscript. Below, we provide a detailed response to each point raised and indicate where the revisions were made in the manuscript.

**Reviewer #1 Comments and Author Responses:**

**Comment:** Please revise the sentence on Lines 112–113 to ensure citation format compliance.

*Response:* The sentence has been corrected as requested to: "The viral concentration (number of viral copies) was determined based on the standard curve as described by Curcio et al. 2022. The citation formatting was updated according to journal guidelines.

Location: Methods Section — Lines 112–113

**Comment:** Replace "MAYV standard curve as described by (21)" in the Figure 1 legend.

*Response:* The phrase has been replaced with: "MAYV standard curve as described previously by Curcio et al. 2022.

Location: Page 33 — Figure 1 Legend

**Comment:** Ensure consistency of past tense in procedural descriptions.

*Response:* All procedural sentences in the Methods section were reviewed and revised to maintain past tense consistently.

We hope that the revisions meet the expectations of the Editor and Reviewer. We remain at your disposal for any further modifications that may be needed.

Sincerely,

Juliana Santana de Curcio